# Design and Validation of an Obstacle Contact Sensor for Aerial Robots

**DOI:** 10.3390/s24237814

**Published:** 2024-12-06

**Authors:** Victor Vigara-Puche, Manuel J. Fernandez-Gonzalez, Matteo Fumagalli

**Affiliations:** Department of Electrical and Photonics Engineering Automation and Control, Technical University of Denmark, Elektrovej, 2800 Kongens Lyngby, Denmark; mjfgo@dtu.dk (M.J.F.-G.); mafum@dtu.dk (M.F.)

**Keywords:** aerial systems “mechanics and control”, aerial systems “perception and autonomy”, AI-enabled robotics, force and tactile sensing, motion and path planning, soft sensors and actuators

## Abstract

Obstacle contact detection is not commonly employed in autonomous robots, which mainly depend on avoidance algorithms, limiting their effectiveness in cluttered environments. Current contact-detection techniques suffer from blind spots or discretized detection points, and rigid platforms further limit performance by merely detecting the presence of a collision without providing detailed feedback. To address these challenges, we propose an innovative contact sensor design that improves autonomous navigation through physical contact detection. The system features an elastic collision platform integrated with flex sensors to measure displacements during collisions. A neural network-based contact-detection algorithm converts the flex sensor data into actionable contact information. The collision system was validated with collisions through manual flights and autonomous contact-based missions, using sensor feedback for real-time collision recovery. The experimental results demonstrated the system’s capability to accurately detect contact events and estimate collision parameters, even under dynamic conditions. The proposed solution offers a robust approach to improving autonomous navigation in complex environments and provides a solid foundation for future research on contact-based navigation systems.

## 1. Introduction

In aerial robotics, detecting and recovering from a collision is critical, especially for drones operating in complex, cluttered environments. Despite advances in autonomous navigation [1], significant challenges remain in localization, mapping, and obstacle detection. Failures in these areas often lead to collisions, underscoring the importance of contact sensing as a complementary strategy to ensure resilience during impacts.

Cluttered environments, such as dense forests, pose significant challenges to autonomous navigation systems due to their complex and unpredictable nature. Localization systems, such as Visual Inertial Odometry (VIO) [2] and Simultaneous Localization and Mapping (SLAM) [3], can suffer from drift in GNSS-denied environments, leading to navigation errors. For example, in dense forests, VIO systems combined with LiDAR-based SLAM have been shown to experience significant drift, causing substantial navigation errors and eventual collisions [4]. Additionally, perception systems face difficulties in accurately detecting obstacles within these environments. Mapping systems relying on stereo cameras or LiDAR often struggle with small or thin obstacles, such as leafless branches, as highlighted in [5], where these limitations led to multiple test flight failures. These examples illustrate the compounded challenges of localization and perception in cluttered settings, emphasizing the need for robust collision detection systems to mitigate these failures and enhance resilience in such demanding scenarios.

To address these inevitable collisions, the sense of touch can be implemented as a crucial complementary capability to existing navigation systems. Conventional solutions have been implemented using accelerometers, which can only detect high-impact forces. So, soft collisions could cause the robot to get stuck. It is also possible to detect contacts by comparing the expected motion of the system and the current motion. Rude [6] detected collisions when odometry does not match the expectation. The main advantage of these approaches is that these kinds of sensors are almost always implemented on robots, so no new implementations are needed.

Several sensors can be used to know if a platform is in contact with an obstacle and where the obstacle is. The typical sensors found mainly in ground robots are tactile sensors [7], whiskers [8], and proximity sensors [9].

Regarding whiskers, many solutions can be implemented to simulate these mammals’ hair to sense the environment. One option to simulate whiskers is flex sensors [10]. The working principle of a flex sensor is based on the concept of variable resistance due to mechanical deformation. Papachristos [11] used these sensors as whiskers integrated perimetrically into the robot’s body. Each robotic antennae, as they call it, provides force feedback estimates through bending angles during physical interaction. The interacting antennas are calibrated against force and bending angles. Given the estimates of the interacting forces, they implement a reactive control strategy to ensure collision-tolerant flight. However, this approach of perimetrically distributed antennas would only detect collisions with planar surfaces, so small or thinner objects could collide with the robot platform between each antenna because there are blind areas.

Ponticelli [12] presented another flex sensor approach. Instead of detecting collisions directly with the flex sensors colliding with the obstacles as conducted in the previous approach, flex sensors are used to measure displacements between two platforms. The sensors are placed around the mobile system’s perimeter and connected to an external sliding platform. A fuzzy inference system is implemented to predict the contact zone, classifying the contacts into eight zones.

Briod [13] presented another approach accounting for 3D collision detection. This study uses the AirBurr protective structure [14]. This structure is formed by 24 buckling springs arranged in eight tetrahedral configurations. Hall sensors have replaced eight pivot joints measuring the deformation of low-stiffness springs, providing a force range of 1N. So, they can detect low impacts that accelerometers cannot differentiate between an impact or a normal flight maneuver.

Aucone [15] designed a contact-detection platform based on torque sensors. They equipped a quadrotor with a disc-shaped shell protecting the drone frame from collisions. The shell surface is made of low-friction fiberglass, allowing obstacles to slide along its surface. A six-axis load cell connects the quadrotor’s frame and the shell to enable force feedback control. So, the haptic sensor measures the net wrench resulting from contacts occurring across the entire shell surface. The disc-shaped shell protection and detection mechanism have been tested in push and slide tasks.

Each of the existing contact-detection techniques lacks some information that is valuable for collision recovery. Some solutions provide discretized contact orientation, which causes a loss of resolution, while others rely on rigid platforms that merely detect the presence of a collision without offering detailed feedback. Addressing these deficiencies, this paper proposes an obstacle contact-detection platform designed to complement existing navigation systems.

The proposed platform offers two main advantages. First, it continuously monitors collisions along the entire perimeter without discretization, achieving higher precision in collision detection. Second, it provides real-time feedback by measuring the displacement of the outer ring relative to the fixed inner ring during a collision event up to a range of 2.5 cm. The system can complement vision-based navigation, which can fail in conditions like fog, low light, or smoke. For example, during source-seeking in firefighting, a drone navigating a smoke-filled room can rely on the contact-detection platform to sense and respond to obstacles when vision is impaired. This ensures safe and effective operation, overcoming the limitations of vision-based systems.

The contents of this paper are as follows: Section 2 explains the development of the contact-detection platform, dataset creation, contact prediction algorithms development, and the development of a contact-based autonomous navigation stack. Section 3 describes the experimental study, and the results from different experiments are presented. Finally, the conclusions drawn from this study are summarized in Section 4.

## 2. Methodology

### 2.1. Structure Design and Prototype Development

In this paper, an obstacle contact-detection platform for UAVs is designed as shown in Figure 1. The design was developed to focus on the output variables needed from the contact-detection algorithm. The main variable to predict is the presence of contact. However, this information alone may not be enough to recover from a collision. Understanding the contact orientation and the real-time collision feedback is equally critical. This highlights the importance of our approach in creating a more reliable system for collision recovery.

The platform comprises four key components: the inner platform, outer platform, sensor integration unit, and an elastic band mechanism.

The inner platform consists of two components. Eight inner ring arms, light blue pieces in Figure 1, are attached perpendicularly to the drone’s arm frame via screw connections, forming a square structure. There are four inner ring supports, dark blue pieces in Figure 1, attached with a screwed joint to a pair of inner ring arms, forming a protective perimeter around the propellers, containing the sensor placements and a sliding slot enabling radial movement of the outer ring.

The outer platform comprises two main parts. The outer ring sensor part, light green pieces in Figure 1, contains the sensor tip and sliding support, which enables the radial displacements between the inner and outer platform when inserted into the sliding slot. The outer ring non-sensor part, dark green pieces in Figure 1, completes the full outer perimeter, and both parts are joined using a snap-fit joint, grey pieces in Figure 1.

Then, both platforms are connected by an elastic band mechanism, which applies a force to keep the outer ring centered relative to the inner ring when no contact is applied to the platform. This mechanism is assembled with a pair of attachments, orange pieces in Figure 1, for each sliding support, tensioning an elastic band. This mechanism does not prevent the outer ring from rotating due to non-radial forces. However, despite this limitation, the platform’s simplicity and low cost have demonstrated accurate results during testing.

The sensors used to measure relative displacements between both platforms are flex sensors. Each sensor measures displacement along its longitudinal axis. Therefore, at least two perpendicular sensors must capture displacement measurements in the 2D space. However, a configuration of four sensors is preferred for redundancy and to ensure symmetrical data collection. The sensor tips are protected with heat-shrinkable shields and fixed in their designated slots on both platforms using silicone. The sensors are installed in a pre-bent position, as shown in Figure 2, to leverage the detection of outer ring displacements in opposite directions.

During a collision, the outer ring platform responds to the impact from an obstacle, resulting in a radial displacement of the outer platform relative to the inner platform in the direction of the impact. This displacement is measured by the flex sensors located between both platforms. The sensor data are then fed into the contact-detection algorithm, which detects the collision and provides its orientation and displacement magnitude.

A prototype is developed according to the designed structure of the obstacle contact-detection platform. All the parts described above have been 3D printed using PLA material. The flex sensors used are from Spectra Symbol [10]. Figure 2 shows the developed obstacle contact-detection platform. The radial displacement range is 2.5 cm.

### 2.2. Data Acquisition and Dataset Creation

The implementation for data acquisition consists primarily of an Arduino microcontroller integrated into the LattePanda platform, which is used as the on-board computer attached to one of the legs of the drone body as shown in Figure 2. This microcontroller captures and processes analog signals from the flex sensors.

Each flex sensor is connected with a voltage divider configuration to the LattePanda computer, as shown in Figure 3. The voltage divider is located in the drone body, while the flex sensors are located within the collision platform as shown in Figure 2. One end of the flex sensor is connected to the positive voltage supply, and the other is connected with a resistor in series to the ground. The point between the flex sensor and the resistor serves as the output of the voltage divider, which is fed into the Arduino’s analog inputs.

Under software control, the Arduino’s built-in Analog-to-Digital Converter (ADC) reads these analog inputs sequentially and converts them into digital values. Then, a developed Python script on the LattePanda reads these digital values from the Arduino via a serial connection at a 100 Hz frequency. The script is responsible for further processing and analyzing the data collected from the flex sensors.

Given the data-acquisition system, two data-recording methods were developed to obtain contact information from flex sensors. In both cases, recordings are saved within a CSV file as shown in Table 1. The first four columns represent the raw sensor measurements. Then, there are three labels for each measurement: *collision_pred* (boolean indicating if there is a collision), *angle_gt* (collision orientation in degrees), and *displacement_gt* (collision displacement in centimetres).

The first method consists of recording contact information using a calibrator tool in a static environment, as shown in Figure 4. This tool can record different contact orientations (0°, 45°, 90°, 135°, 180°, 225°, 270°, 315°) and displacements (0.5 cm, 1 cm, 1.5 cm, 2 cm, 2.5 cm), as shown in Figure 5.

The calibration dataset (C) is created using the calibrator tool method. In addition, static and flight data without collisions are recorded and added to this calibration dataset to contain contact and no contact information. This dataset aims to train the initial contact-detection model, which is used later to label contact events for the flight dataset.

The second method employs an OptiTrack system to track manual flight collisions. Here, three OptiTrack markers define the wall collision plane, while additional markers track the drone center. The contact orientation is derived from the angular difference between the drone’s yaw vector and the wall’s normal vector. The contact displacement is calculated by measuring the initial contact distance and tracking the drone’s changing distance during the collision. Continuous data on the drone’s position are recorded until the collision ends, as illustrated in Figure 6. As there was no methodology to detect contact events with the OptiTrack system, the collision labels correspond to collision predictions from a model trained previously with the calibration dataset. Eleven manual flights were recorded using the OptiTrack system. Each flight contains around 20 collisions distributed around the drone, as Figure 6 illustrates. Two datasets are created with the flight data.

On the one hand, the Flight Discrete Dataset (FD) is designed for training models using discrete contact samples. Each sample represents a single time step, as shown in Table 1.

On the other hand, the Flight Time-Series Windows Dataset (FW) is tailored for training models with time-series information. Each training sample consists of a sequence of the most recent measurements within a defined window size. This dataset provides the model with contextual information about the sensors’ behavior over time. The label for each time-series sample corresponds to the current time step within the window.

We used nine flights to train the model for the FD and FW datasets, while the remaining two were reserved for evaluation. Due to the predominance of non-contact data in the training set, the dataset was balanced to ensure an equal number of contact and non-contact samples.

### 2.3. Model Construction

Flex sensors inherently exhibit nonlinear behavior due to the variability in their temperature resistance coefficient with changing operating temperatures. As shown in Figure 5, a simple logic can be applied to predict contact when force is aligned with specific sensor orientations (0°, 90°, 180°, 270°) as the corresponding sensor’s measurement increases. However, when applying forces at intermediate orientations (45°, 135°, 225°, 315°), the platform design induces torsional movements to the flex sensors, further contributing to the sensor’s non-linearity, resulting in inconsistent measurement decreases.

To address this non-linearity, Neural Networks (NNs) were chosen as the algorithm to predict contact information from the flex sensor data. Two different tasks are handled using this data: a binary classifier to distinguish between contact events and a continuous regression network to predict the contact orientation and displacement.

The NNs have been trained with different input data. On the one hand, using the differences between each sensor (D) intends to normalize sensor measurements for different temperature conditions. On the other hand, raw sensor measurements (R) have also been used to train the models.

Both models have been trained using a learning rate of 0.001 and a batch size of 32.

#### 2.3.1. Contact-Detection Model

A Feed Forward Neural Network (FFNN) has been implemented to predict contact events. It consists of the input layer adapted to each input type, one hidden layer containing 16 units, followed by ReLU activation functions to introduce non-linearity, allowing it to learn different complex patterns. The output layer uses a Sigmoid activation function mapping the input values to the range [0, 1], representing the probability that the predictions belong to one class or the other. Finally, the Binary Cross Entropy Loss function trains the model, detailed in Equation (Equation 1).
(1)$$\mathrm{Binary}\;\mathrm{Cross-Entropy}=-\frac{1}{n}\sum_{i=1}^{n}\left[y_{i}\log\left({\hat{y}}_{i}\right)+\left(1-y_{i}\right)\log\left(1-{\hat{y}}_{i}\right)\right]$$
where $y_{i}$ is the actual label and ${\hat{y}}_{i}$ is the predicted probability.

#### 2.3.2. Angle and Displacement Model

For this task, two different architectures have been proposed: an FFNN (F) and a combination of FFNN with Long-Short Term Memory (LSTM) blocks (L+F) to leverage time-series information, shown in Figure 7.

The first approach, shown in Figure 7a, contains one general hidden layer with 16 units that extracts features relevant to angle and displacement predictions. Subsequently, three sets of one hidden layer, with 32 units each, are specialized in predicting collision angle sine, angle cosine, and displacement. A ReLU activation function follows each hidden layer.

For the second approach, shown in Figure 7b, the NN input is a matrix containing the time series from all the sensors instead of discrete sensor measurements, where the window size (W) is an experiment parameter. The general hidden layer from the first approach is substituted by an LSTM block containing one hidden layer and a hidden size 32. Then, the LSTM block diverges again into three sets of one hidden layer, with 32 units each, specialized in predicting collision angle sine, angle cosine, and displacement. A ReLU activation function follows each hidden layer.

Finally, both architectures have a linear output layer without any activation function, allowing the network to produce continuous output values directly corresponding to the predicted angles (sine and cosine) and displacements, as shown in Figure 7.

The training process employs a combined loss function to optimize the model. The displacement loss is calculated using the Mean Squared Error (MSE).
(2)$${\mathrm{MSE}}_{\mathrm{displacement}}=\frac{1}{N}\sum_{i=1}^{N}{\left(y_{i}-{\hat{y}}_{i}\right)}^{2}$$
where $y_{i}$ is the *i*-th actual value and ${\hat{y}}_{i}$ is the *i*-th predicted value.

The angle loss is determined using a custom Sine Cosine Distance loss, as detailed in Equation (Equation 3).
(3)$$\mathrm{Sine}\;\mathrm{Cosine}\;{\mathrm{Distance}}_{\mathrm{angle}}=\sqrt{{\left(\sin\left(\hat{\theta }\right)-\sin\left(\theta \right)\right)}^{2}+{\left(\cos\left(\hat{\theta }\right)-\cos\left(\theta \right)\right)}^{2}}$$
where $\sin(\hat{\theta })$ and $\cos(\hat{\theta })$ are the predicted sine and cosine values, and sin(θ) and cos(θ) are the target sine and cosine values.

The final loss function is the sum of these two components, detailed in Equation (Equation 4).
(4)$$\mathrm{Angle}\;\mathrm{Displacement}\;\mathrm{Loss}={\mathrm{MSE}}_{\mathrm{displacement}}+\mathrm{Sine}\;\mathrm{Cosine}\;{\mathrm{Distance}}_{\mathrm{angle}}$$

### 2.4. Autonomous Contact-Based Navigation

The autonomous stack shown in Figure 8 has been created to provide insights about the collision platform performance when integrated into a real autonomous mission. The contact predicted information will feed the system to enable collision recovery during an autonomous mission.

The drone is equipped with the developed collision platform and a LiDAR Livox Mid360 to map the environment, as shown in Figure 2. The localization will be based on an OptyTtrack system that provides the drone’s position and orientation.

As the mapping subsystem is not the focus of this project, the existing ROS2 package *Octomap Server* [16] has been used to create the 3D grid map based on the point cloud generated by the LiDAR sensor. The map resolution has been set to 0.88 m as the collision platform diameter.

The path-planning algorithm used is the RRT*, which is supported by environment information provided by the mapping subsystem in the form of a 3D grid map.

The parameters from the planning algorithm are explained in Table 2.

The logic created for the planning algorithm is shown in Figure 9.

Every time the path-planning node is launched, it is initialized by calculating the initial path from the starting point to the goal coordinates using the RRT* algorithm. If the 3D grid map is available, it will be used to plan a free-obstacle trajectory.

Once initialized, the following logic is executed at a fixed rate. When a new 3D grid map is received from the mapping node, the *HANDLE OCTOMAP* submodule verifies if any obstacles intersect with the previously calculated path. If such collisions are detected, the submodule recalculates an obstacle-free path using the RRT* algorithm from the current drone position, incorporating the latest obstacle information.

When a contact-detection message is received, the *HANDLE COLLISION DETECTION* module handles the collision, guides the drone to a collision-free state, and recalculates a new path. The collision response logic involves returning to previous waypoints within a no-contact state.

Here is how the collision handling process works:**Collision Detection**: Upon detecting a collision, the target waypoint is immediately decremented by the *Step Back* parameter, directing the drone to move backward to the previous waypoints. At this point, the drone is in a collision-recovery state.**Collision Recovery**: In the collision-recovery state, there are two possible scenarios:**Persistent Contact**: If the drone continues to detect contact while moving backward, the target waypoint will keep decreasing, sending the drone further back to previous waypoints until a no-contact state is reached.**No-contact State**: The collision state is finished once the drone moves to a waypoint where no contact is detected.**Post-Collision Handling**: After resolving the contact event, which involves detecting the collision, recovering from it, and reaching a no-contact state, the detected contact is saved on the map using the initial predicted angle. The obstacle world frame position is calculated as follows:
(5)$$o^{uav}=\left[\begin{array}{c} \cos\left(\theta_{c}\right)\cdot r_{d}\\ \sin\left(\theta_{c}\right)\cdot r_{d}\\ 0 \end{array}\right]$$(6)$$R_{uav}^{w}=\left[\begin{array}{ccc} \cos(\psi ) & -\sin(\psi ) & 0\\ \sin(\psi ) & \cos(\psi ) & 0\\ 0 & 0 & 1 \end{array}\right],$$(7)$$o^{w}=R_{uav}^{w}\cdot o^{uav}+d^{w}$$
where $\theta_{c}$ is the collision angle in degrees, $r_{d}$ is the radius of the drone, $o^{uav}\in R^{3}$ is the position of the obstacle in the drone frame, $d^{w}\in R^{3}$ is the position of the drone in the world frame, $R_{uav}^{w}\in R^{3\times 3}$ is the rotation matrix from the drone frame to the world frame based on the yaw angle of the drone, ψ is the yaw angle of the drone in radians, and $o^{w}\;\in R^{3}$ is the position of the obstacle in the world frame.Subsequently, a new path is calculated to avoid the obstacle that caused the collision. Whenever a collision is received,

Finally, the *DECIDE TARGET WAYPOINT* module sends the target waypoint to the offboard control node. In the normal operation, without collisions, the target waypoint is increased when it is reached. In case of a collision, the target waypoint is selected based on the logic explained above by the *HANDLE COLLISION DETECTION* submodule.

## 3. Experimental Study

An experimental evaluation was performed to assess the accuracy of the obstacle contact-detection platform. This process included dataset creation, model construction, and results analysis.

### 3.1. Contact-Detection Model

The results from all the contact-detection models evaluated with the flight datasets are shown in Table 3.

As detailed above, it is crucial to note that Model 1, trained with calibration data, was used to label the contact events in the flight dataset. That is why Model 1 exhibits high accuracy and precision values. However, it also showed a lower recall, indicating that the model predicted too many false negatives and missed many contact-labeled events.

Models trained using the differences in sensor values (Models 2 and 4) as input features for the models generally perform poorly. This codification seems to be losing valuable information for the model to understand the collisions.

Conversely, Model 3, trained with raw sensor values as inputs and contacts from the flight dataset, outperformed all metrics. This fact indicates its ability to predict contact events accurately and avoid mispredictions. Model 3 training loss is shown in Figure 10a for 100 epochs. The mean processing time of the Model 3 is 0.28 ms, which is significantly faster than the measurement update rate of 10 ms, ensuring that the model can process sensor data in real time without delay.

These results underscore the critical importance of training models with data resembling real-world scenarios. This approach enables the model to generalize effectively, a critical factor in its performance.

### 3.2. Angle and Displacement Model

The results from all the angle and displacement models evaluated with the flight datasets are shown in Table 4.

It is clear that models trained solely with calibration data (Models 1 and 2), without flight information, struggle to generalize when evaluated with flight datasets. This fact underscores the crucial importance of training with data closely resembling real conditions.

The same pattern analyzed previously for the contact-detection model is present for the angle and displacement model. When training the model with sensor differences as inputs (Models 2 and 4), it performs worse than the ones trained with discrete raw values or raw values within time series.

A clear trend regarding angle predictions has been discovered in how they perform better by increasing the model complexity by adding the LSTM block (Architecture L+F) against the simple FFNN (Architecture F).

Remarkably, models incorporating time-series information (Models 5, 6, and 7) substantially improve angle prediction. These models achieve a near 50% reduction in the predicted angle errors such as MAE, MSE, and RMSE. Among the time-series trained models, windows size 20 and 30 seem to perform better than windows size 10, with which not enough information could be captured.

However, angle and displacement predictions have demonstrated that they follow different trends for model design. As seen in Table 4, the best displacement prediction model is Model 3, the simplest one trained with flight data. This fact contradicts the trend in angle prediction, where increasing model complexity led to better results. These unexpected findings challenge our assumptions of creating a combined neural network to predict angles and displacements leveraging shared features.

Model 6 stands out in correctly predicting both angles and displacements. Model 6 training loss is shown in Figure 10b for 300 epochs. The mean processing time of Model 6 is 2.54 ms, which is significantly faster than the measurement update rate of 10 ms, ensuring that the model can process sensor data in real-time without delay. Figure 11 illustrates the comparison between the true and predicted angles and displacements derived from the flight test dataset by using Model 6.

Figure 11a shows how the model can almost fit the predicted versus true collision angles into a line along the entire UAV perimeter, with just a few visible outliers. This means that no significant errors are present. It can also be noticed that for angles 0 and 360, predictions of 360 and 0, respectively, are conducted, not being a problem because of the circular nature of the angles.

However, Figure 11b demonstrates that displacement prediction is not working as well as the angle prediction. A trend can be seen in the predicted versus true displacements that fit the red dashed line (representing the best model performance), but the predictions are more dispersed than the angle predictions.

Regarding the prediction during the contact events, from the initial contact to the contact release, Figure 12 and Figure 13 leverage the time-series nature of a contact event.

On the one hand, Figure 12 shows how the angle prediction evolves during the contact event. Figure 12a shows the predicted angle oscillates around the true angle due to the sensor values’ abrupt changes from one sample to another, leading to abrupt predictions but still around the true label with an error below 5 degrees. Figure 12b shows a more significant angle prediction error of 50 degrees at the beginning, decreasing until 30 degrees at the end of the contact event.

On the other hand, Figure 13 shows how the displacement prediction evolves during the contact event. Figure 13a shows how the predicted displacement imitates the displacement pattern from the true label, demonstrating its potential to predict continuous displacements. However, Figure 13b shows the correct predicted displacement pattern, increasing and then decreasing but with a clear offset to the true displacement.

### 3.3. Autonomous Contact-Based Navigation

In this experiment, the navigation stack has to reach a target waypoint. A goal-shaped obstacle (2 × 3 m) is deliberately placed along this path. This goal’s left half is obstructed by thin tensioned vertical wires, marked in red in Figure 14, which the LiDAR sensor cannot detect. The objective is to demonstrate how the collision platform can compensate for the LiDAR’s lack of resolution, so the initial trajectory will be defined to send the drone against the wires, green trajectory in Figure 14b, while the LiDAR maps the environment, white boxes in Figure 14b. The mission visualization is shown in Figure 14b, where the red arrow defines the drone position, and the collision platform diameter is defined by the grey circle centered in the arrow.

The mission visualization result is shown in Figure 15. Figure 15a shows the initial trajectory that goes through the wires not detected by the LiDAR. Then, Figure 15b shows the collision event detected and included in the map as a green box. Figure 15c shows the collision recovery maneuver by moving backward to a non-contact state when the collision is detected. Finally, Figure 15d shows the new recalculated path avoiding LiDAR and contact-based detected obstacles and the drone avoiding the wires and moving to the goal.

This experiment demonstrates how the path-planning process integrates LiDAR and contact information to enhance obstacle detection and avoidance. The LiDAR sensor provides a 3D map of the environment it can detect, while the contact-detection system refines this map by adding obstacles not detected by the LiDAR. This combination results in a more comprehensive environmental model.

Regarding the mission metrics, shown in Table 5, the LiDAR and contact-based navigation missions have been completed with only one collision. The duration of the contact event was 1.053 s. This extended duration is likely due to the flexible wires, resulting in a smoother and less abrupt collision as the drone pushes against them. This highlights the platform’s ability to detect flexible collisions effectively.

The reaction time was 0.538 s. This slow reaction may be attributed to integrating LiDAR data into the planning algorithm, which can introduce additional computational delays.

The prediction angle error was 3.54 degrees, critical in accurately mapping the collision. This low error margin ensured that the collision was precisely included in the map, allowing the drone to replan its path effectively and avoid the obstacle with just one collision.

## 4. Conclusions

This paper proposed a novel obstacle contact-detection platform, which can be integrated into an existing drone frame to obtain collision information. The following conclusions are obtained:

(1) The obstacle contact-detection platform is designed, and the prototype is developed. Flex sensors are integrated between the inner and outer platforms. When the outer platform collides, collision detection can be realized based on the flex sensor displacements between platforms.

(2) A contact-detection algorithm is developed based on neural networks that provide contact detection, orientation, and displacement. This algorithm runs on the onboard computer.

(3) The experimental system of the obstacle contact-detection platform is built, and the contact-detection strategy is verified through the actual experimental test. The contact-detection results show an accuracy of 0.99. Regarding angle prediction, an MAE of 9.31° with an SD of 10.84°, and regarding displacement prediction, an MAE of 0.58cm with an SD of 0.79cm are achieved. In addition, autonomous contact-based navigation has been achieved by using the predicted contact information, compensating for the LiDAR’s lack of resolution.

The obstacle contact-detection platform, with its comprehensive perimeter contact information and real-time collision feedback, enhances the UAV’s ability to recover from collisions and continue its mission. This research introduces a novel approach to contact sensing, which has the potential to inspire and shape future developments in collision-tolerant robotics, offering hope for a future with reduced risk of autonomous mission failures.

## Figures and Tables

**Figure 1 sensors-24-07814-f001:**
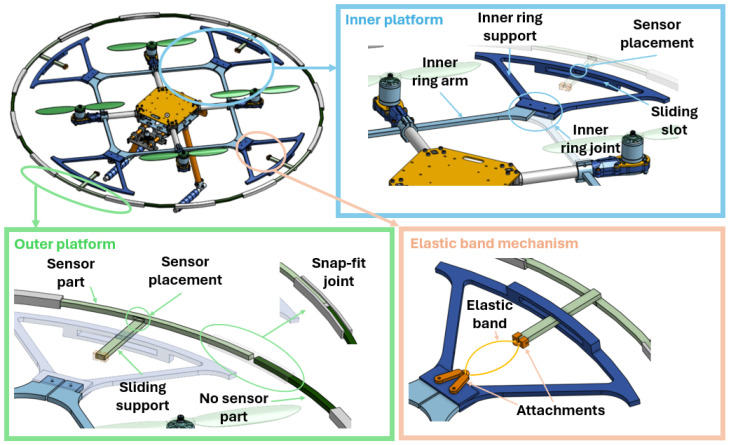
Overall structure of obstacle contact-detection platform.

**Figure 2 sensors-24-07814-f002:**
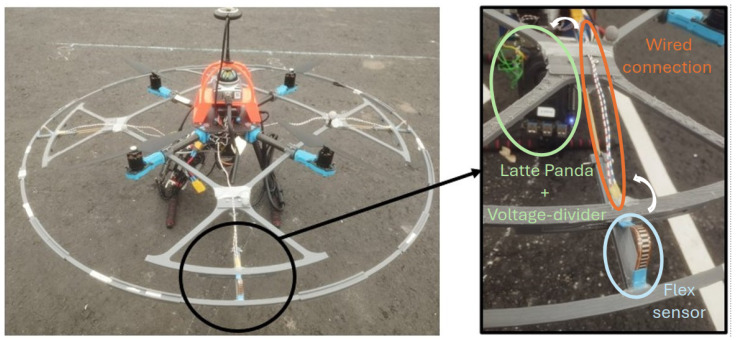
Developed prototype of obstacle contact-detection platform.

**Figure 3 sensors-24-07814-f003:**
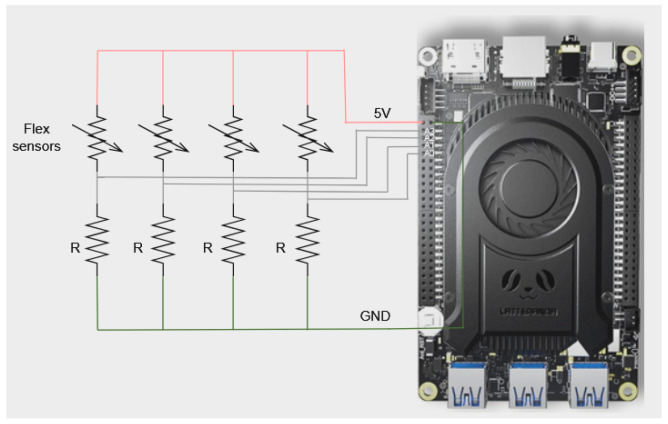
Flex sensor voltage divider schematic.

**Figure 4 sensors-24-07814-f004:**
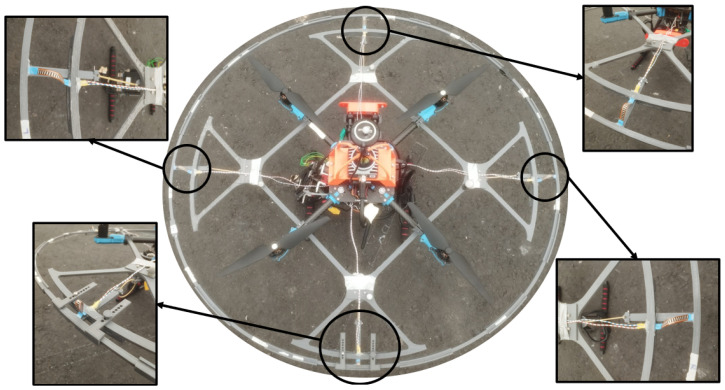
Collision platform with calibrator tool: 180° orientation/2.5 cm displacement.

**Figure 5 sensors-24-07814-f005:**
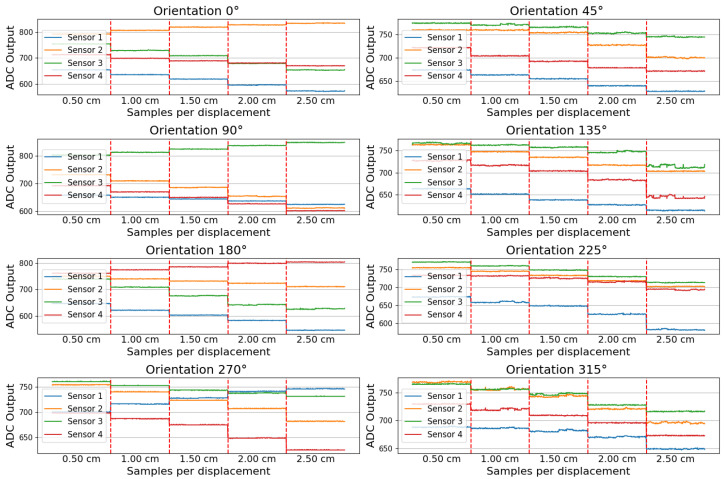
Calibrator tool data recordings. Each plot corresponds to a different contact orientation. For each orientation, the following displacement sections between red vertical dashed lines are shown: 0.5, 1, 1.5, 2, and 2.5 cm.

**Figure 6 sensors-24-07814-f006:**
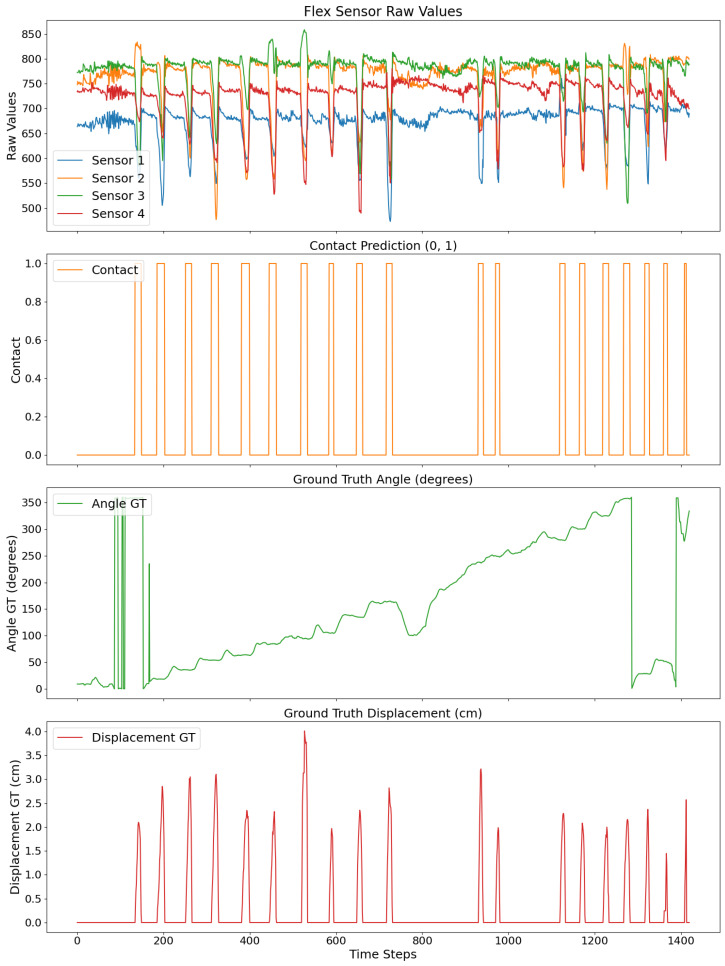
OptiTrack data recording corresponding to one flight. The first plot shows flex sensor raw values for different collisions during a flight. The second plot shows the contact events predicted. The third plot shows the drone’s orientation during the flight. The fourth plot shows the contact displacements of the outer platform recorded for each collision event.

**Figure 7 sensors-24-07814-f007:**
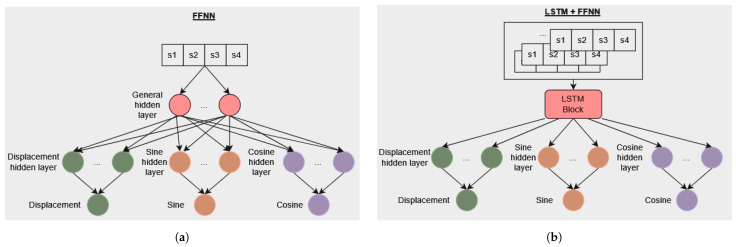
(**a**) Angle and displacement FFNN model (F). (**b**) Angle and displacement LSTM+FFNN model (L+F).

**Figure 8 sensors-24-07814-f008:**
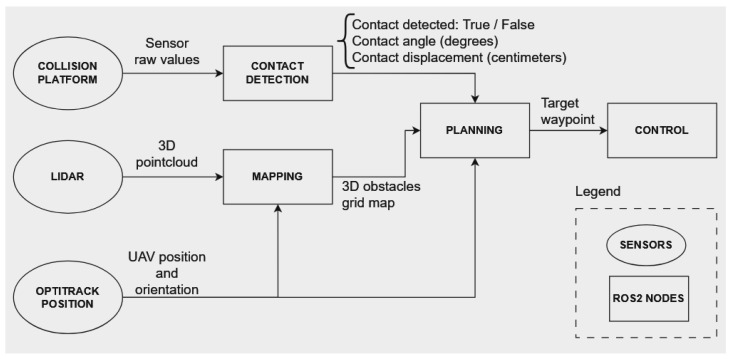
Autonomous pipeline proposed to solve contact-based navigation.

**Figure 9 sensors-24-07814-f009:**
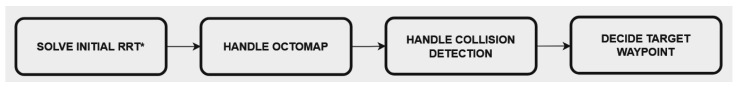
Path-planning submodules proposed combining LiDAR 3D grid map and contact information for obstacle avoidance.

**Figure 10 sensors-24-07814-f010:**
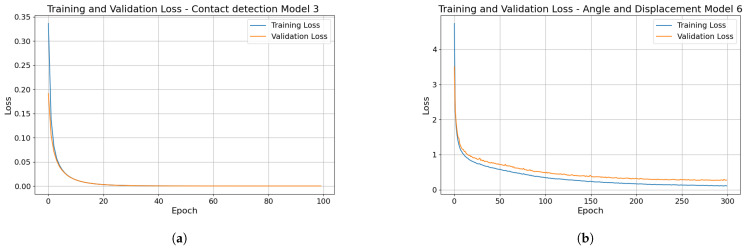
(**a**) Training and validation loss of the contact-detection Model 3. (**b**) Training loss of the angle and displacement Model 6.

**Figure 11 sensors-24-07814-f011:**
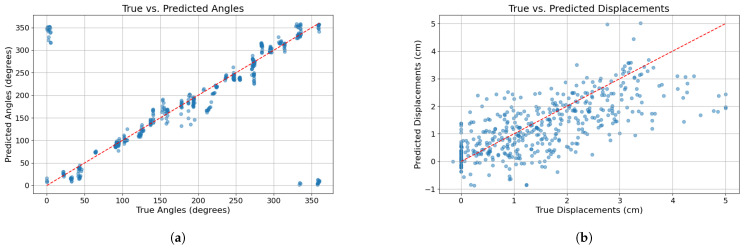
(**a**) Model 6 true versus predicted angles test flight dataset. (**b**) Model 6 true versus predicted displacements test flight dataset.

**Figure 12 sensors-24-07814-f012:**
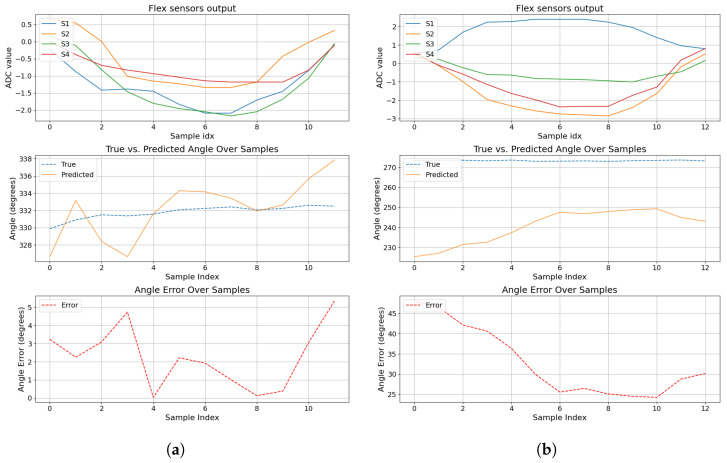
Contact event plot over time using Model 6. The first plot shows flex sensor values. The second plot shows the true angle versus the prediction. The last plot shows the angle error during the collision. (**a**) Contact event at 332°. (**b**) Contact event at 270°.

**Figure 13 sensors-24-07814-f013:**
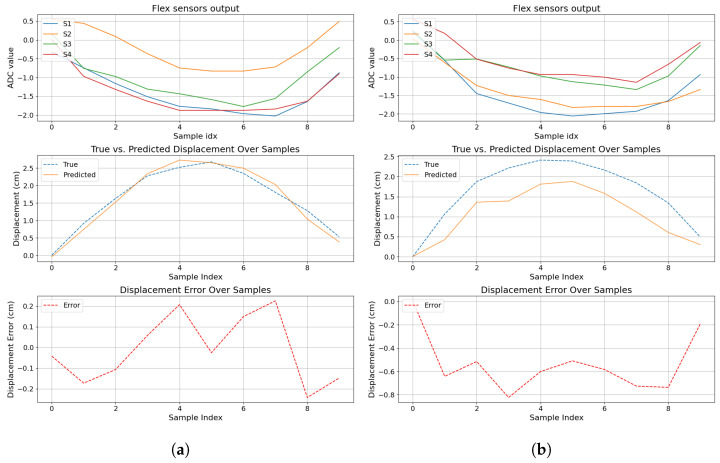
Contact event plot over time using Model 6. The first plot shows flex sensor values. The second plot shows the true displacement versus the prediction. The last plot shows the displacement error during the collision. (**a**) Contact event 1. (**b**) Contact event 2.

**Figure 14 sensors-24-07814-f014:**
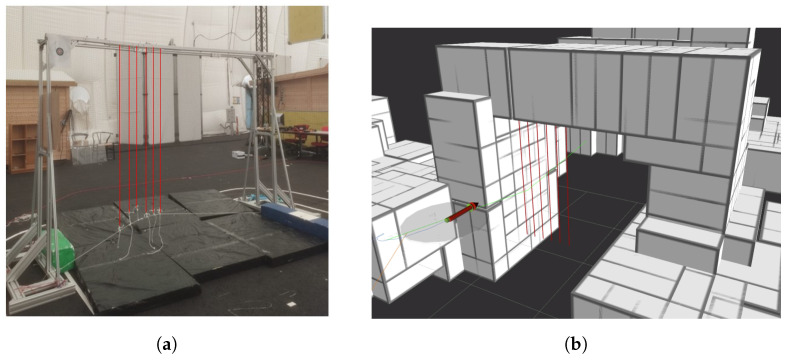
Lidar and contact-based navigation experiment scenario. (**a**) Real scenario; (**b**) visualization scenario. The red arrow indicates the drone’s position and orientation, the grey circle centered at the red arrow’s origin marks the collision platform’s outer perimeter, the green line depicts the drone’s planned flight path, and the red vertical lines signify the thin wire obstacles.

**Figure 15 sensors-24-07814-f015:**
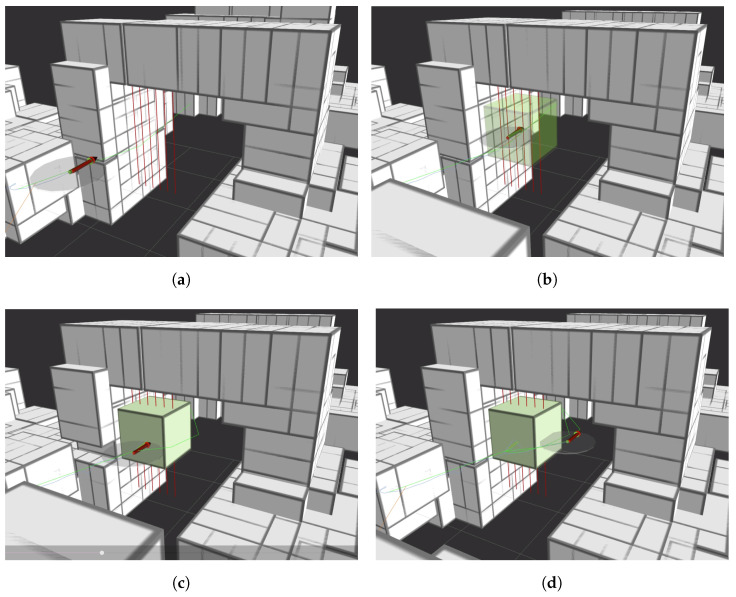
Lidar and contact-based navigation experiment showing planning and mapping stages. (**a**) Initial path with octomap; (**b**) collision and contact map updated; (**c**) collision recovering; (**d**) recalculated path.

**Table 1 sensors-24-07814-t001:** Contact data recordings data structure from one sample.

Sensor1	Sensor2	Sensor3	Sensor4	collision_pred	angle_gt	displacement_gt
672.5	726.29	829.90	709.70	1	87.42	1.07

**Table 2 sensors-24-07814-t002:** Path-planning parameters.

Parameters	Explanation
Contact map resolution	Square box side length map contact obstacle [m]
Octomap resolution	Square box side length map LiDAR obstacle [m]
RRT Search Space Range	3D volume to search for planning solutions [m]
RRT q	Distance between RRT waypoints [m]
RRT Goal	Mission goal waypoint coordinates [m]
Step Back	Waypoints number to recover from collision [m]

**Table 3 sensors-24-07814-t003:** Contact-detection experiments.

Model	1	2	3	4
Dataset	C	C	FD	FD
Input	R	D	R	D
Accuracy	0.99	0.69	0.99	0.95
Precision	0.98	0.11	0.82	0.47
Recall	0.71	0.93	0.93	0.92
F1 Score	0.82	0.19	0.87	0.62
AUC	0.85	0.8	0.96	0.94

**Table 4 sensors-24-07814-t004:** Angle and displacement model experiments.

Model	1	2	3	4	5	6	7
Dataset	C	C	FD	FD	FW	FW	FW
Architecture	F	F	F	F	L+F	L+F	L+F
Input	R	D	R	D	W	W	W
W size	-	-	-	-	10	20	30
Angle MAE	28.91	43.81	16.61	19.37	12.69	9.64	9.31
Angle MSE	2042.13	3924.93	804.91	1049.96	356.01	190.08	204.23
Angle RMSE	45.19	62.65	28.37	32.40	18.87	13.79	14.29
Angle StdDev	34.73	44.78	23.00	25.97	13.96	9.86	10.84
Angle Median	15.49	22.02	6.52	8.80	7.34	6.69	5.41
Angle Max	177.42	178.90	132.43	177.34	64.22	55.21	59.32
Displacement MAE	1.04	1.16	0.58	0.75	0.64	0.64	0.66
Displacement MSE	1.74	2.78	0.63	0.92	0.71	0.72	0.80
Displacement RMSE	1.32	1.67	0.80	0.96	0.84	0.85	0.89
Displacement StdDev	0.85	1.67	0.79	0.96	0.82	0.83	0.88
Displacement Median	0.94	0.80	0.46	0.63	0.51	0.50	0.51
Displacement Max	4.14	8.08	4.05	4.39	3.68	3.13	3.97

**Table 5 sensors-24-07814-t005:** Lidar and contact-based navigation real experiment results.

Metrics	Results
Number of collisions	1
Contact event duration (s)	1.053
Reaction time (s)	0.538
Angle predicted	3.74
Angle ground truth	0.23
Angle error	3.54

## Data Availability

The data in this study are available upon request from the corresponding author.

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
