# Peer review of "Design and Validation of an Obstacle Contact Sensor for Aerial Robots"

_sensors, 2024, doi:10.3390/s24237814_

Round 1

Reviewer 1 Report

Comments and Suggestions for Authors

The proposed research presents an innovative contact sensor designed to enhance autonomous navigation for aerial robots. While the concept is promising, several areas require further elaboration. The introduction should provide a more comprehensive discussion of existing literature, particularly comparing your approach with traditional obstacle avoidance algorithms to highlight strengths and weaknesses. Clarity is needed regarding the connection method between the sensor and the drone, including details about data transmission (wired vs. wireless) and update frequency. Exploring the applicability of the sensor in source seeking tasks, such as ensuring safe control for quadrotors, would broaden the impact of your research. Additionally, in Equation 5, the dimension and definition of the rotation matrix should be specified for clarity. Lastly, the manuscript would benefit from a more extensive reference list that includes classic and recent studies, providing a stronger foundation for your research. Overall, addressing these points will enhance the manuscript’s contribution to the field of autonomous robotics.

  • Insufficient Discussion of Existing Results: The introduction lacks a thorough discussion of existing results. A comparative analysis with traditional obstacle avoidance algorithms would strengthen the context. Specifically, please include a discussion on the advantages and disadvantages of your proposed approach in relation to these conventional methods.

  • Connection Method Between Sensor and Drone: Clarify the connection method between the sensor and the drone. How is the data transmitted (wired or wireless)? Additionally, what is the data update frequency? Can the system support wireless transmission? These details are crucial for understanding the practical implementation of your sensor.

  • Applicability in Source Seeking: Discuss whether the sensor can be employed in source seeking tasks for quadrotors, particularly in ensuring safe control (e.g., source-seeking for quadrotors). This application could significantly enhance the relevance and versatility of your sensor design.

  • Dimension and Definition of Rotation Matrix: In Equation 5, please specify the dimension of the rotation matrix. Providing a clear definition and context for this equation will enhance the comprehensibility of your work and ensure that readers can accurately interpret the mathematical formulations.

  • Limited References: The number of references cited in the manuscript is quite low. The inclusion of additional classic and contemporary works would provide a more comprehensive background and context for your research. This will not only strengthen your arguments but also demonstrate the relevance of your work in the broader field.

Comments on the Quality of English Language

The overall quality of the English language in the manuscript is adequate for conveying the research ideas; however, several areas could benefit from improvement. There are instances of awkward phrasing and grammatical errors that may hinder clarity and flow. A thorough proofreading is recommended to correct these issues and enhance readability. Additionally, some technical terms and concepts could be explained more clearly for a broader audience. Improving the language quality will not only make the manuscript more polished but also ensure that the key contributions of the research are communicated effectively.

Author Response

Comments 1: The introduction lacks a thorough discussion of existing results. A comparative analysis with traditional obstacle avoidance algorithms would strengthen the context. Specifically, please include a discussion on the advantages and disadvantages of your proposed approach in relation to these conventional methods.

Response 1: 

Thanks for the feedback on the introduction. We recognize that our initial presentation may not have clearly communicated the motivation and objectives of our work, which might have led to misunderstandings regarding its connection to the relevant literature. To address this, we have revised the introduction, adding a paragraph to better contextualize and clarify our contribution.  

Specifically, we want to emphasize that this work focuses not on obstacle avoidance techniques but on collision detection systems to address failures in key modules such as localization, mapping, and planning. To provide a more precise context, we have incorporated examples of common challenges, such as localization drift in GNSS-denied environments and the inability to detect small obstacles like thin branches, which can lead to collisions and mission failures. By highlighting these limitations, the revised introduction underscores the necessity of a robust collision detection system. While a simple recovery strategy was implemented in this study to demonstrate the detection system's effectiveness, the primary focus remains on detecting collisions rather than developing avoidance strategies.  

In addition, we have presented the main advantages of our contribution after the literature review of the existing contact detection techniques at the end of the introduction.   

Comments 2: Clarify the connection method between the sensor and the drone. How is the data transmitted (wired or wireless)?
Response 2: 
The connection method between the sensor and the drone is detailed in Section 2.2. Specifically, the flex sensor is connected to the Lattepanda onboard computer via a voltage divider, establishing a wired connection. To clarify, the Lattepanda functions as the drone's onboard computer. As illustrated in Figure 3, the wiring between the flex sensor and the Lattepanda is the sole connection required for data transmission. Additionally, we have updated Figure 2 to depict the placement of the sensors, the Voltage-divider and Lattepanda setup, and their wired connection. 

Comments 3: Additionally, what is the data update frequency?
Response 3: Thank you for pointing this out. The data update frequency was not explicitly mentioned in Section 2.2. The sensor data is updated at a frequency of 100 Hz based on readings from the ADC converter integrated into the LattePanda. Additionally, the contact detection model processes each reading with an average time of 0.28 ms, while the angle and displacement model requires an average of 2.54 ms for processing. As a result, all sensor measurements are continuously processed in real-time. We have included the measurements update frequency in Section 2.2 and the model processing time updated in Sections 3.1 and 3.2.

Comments 4: Can the system support wireless transmission? These details are crucial for understanding the practical implementation of your sensor.
Response 4: The collision detection platform in this study is connected to the onboard computer via wires. The wires extend from the flex sensor to the voltage divider and data acquisition elements on the drone body. We have improved our description of the implementation on the text and highlighted it in blue in Section 2.2. In our opinion, wireless transmission between the flex sensors and the latte panda is out of the scope. However, data collected can be wirelessly transmitted to the ground with the current configuration via ROS2 running inside the LattePanda.

Comments 5: Discuss whether the sensor can be employed in source seeking tasks for quadrotors, particularly in ensuring safe control (e.g.,  source-seeking for quadrotors). This application could significantly enhance the relevance and versatility of your sensor design.
Response 5:
We agree that the proposed collision detection platform can be effectively applied in source-seeking missions, particularly in challenging environments. The collision platform can complement vision-based navigation systems, which may be impaired by adverse conditions such as fog, smoke, or poor lighting. For instance, during firefighting missions, a drone navigating through a smoke-filled room could rely on the platform to safely detect and respond to obstacles, ensuring effective source-seeking when vision is compromised. This example demonstrates the sensor’s practical application and highlights its relevance and versatility in enhancing operational safety in demanding scenarios. 

Comments 6: 
In Equation 5, please specify the dimension of the rotation matrix. Providing a clear definition and context for this equation will enhance the comprehensibility of your work and ensure that readers can accurately interpret the mathematical formulations. 

Response 6: We have revised the manuscript to include a clear description of the rotation matrix, specifying its dimensions and relation to the yaw angle. This additional context should improve the mathematical formulation and enhance the clarity for readers interpreting Equation 5. This answer is solved in Equation 6 and the following paragraph.

Comments 7: The number of references cited in the manuscript is quite low. The inclusion of additional classic and contemporary works would provide a more comprehensive background and context for your research. This will not only strengthen your arguments but also demonstrate the relevance of your work in the broader field.
Response 7:
To the best of our knowledge, the references cited in the manuscript represent the most relevant and up-to-date works in the collision detection field that directly align with the focus of our study. 

Reviewer 2 Report

Comments and Suggestions for Authors

In this paper, the authors propose an innovative design for a contact sensor that improves autonomous navigation by enabling effective physical contact detection.  Besides, they developed a neural network-based algorithm for contact detection that can identify contact events, determine orientation, and estimate displacement. Moreover, an experimental system for the obstacle contact detection platform has been constructed, and the contact detection strategy was validated through actual experimental tests. My comments are as follows:

Main issues that need to be modified

1) The introduction should clear articulation of the innovation or unique contribution of this study in relation to existing systems.

2) Although the author claims to be able to absorb impacts, the paper does not evaluate and quantitatively analyze the impact absorption characteristics of the platform.

3) Loss convergence graph of the developed neural network-based algorithm should be given.

Minor issues that need to be modified

4) The font in Figures 5,6,10,11, and 12 is too small, making it difficult for readers to read.

5The format of references needs to be readjusted according to the requirements of the journal to ensure consistency in format.

Author Response

Comments 1: The introduction should clear articulation of the innovation or unique contribution of this study in relation to existing systems.
Response  1: Thank you for your comment. To address your suggestion, we have included two main advantages in the introduction that highlight the unique contributions of this study. First, the system provides continuous collision orientation detection, ensuring higher precision than discretized methods. Second, it offers real-time collision feedback based on the displacement of the outer platform relative to the inner platform, providing detailed information during a collision event. These innovations enhance collision recovery and resilience, distinguishing our system from existing solutions. 

Comments 2: Although the author claims to be able to absorb impacts, the paper does not evaluate and quantitatively analyze the impact absorption characteristics of the platform.
Response 2:
Thank you for your comment. We want to clarify that the term "impact absorption" has been removed from the paper, as no absorption is intended or measured in the platform's design. The research focuses on providing real-time collision feedback based on the collision orientation and displacement of the outer platform relative to the inner platform. The absorption term has been changed in the abstract, Sections 1, 2.1, and 4. 

Comments 3: Loss convergence graph of the developed neural network-based algorithm should be given.
Response 3:
The loss convergence graph for the contact model and the angle and displacement model have been added in Figure 10. 

Comments 4: The font in Figures 5,6,10,11, and 12 is too small, making it difficult for readers to read.
Response 4: 
Figures 5, 6, 10, 11, and 12 have been updated with an appropriate font size to be readable.   

Comments 5: The format of references needs to be readjusted according to the requirements of the journal to ensure consistency in format.
Response 5: The references have been reformatted manually to align with the journal's style guide, ensuring consistency and adherence to the specified requirements.